# Resilience and Mitigation Strategies of Cyanobacteria under Ultraviolet Radiation Stress

**DOI:** 10.3390/ijms241512381

**Published:** 2023-08-03

**Authors:** Varsha K. Singh, Sapana Jha, Palak Rana, Sonal Mishra, Neha Kumari, Suresh C. Singh, Shekhar Anand, Vijay Upadhye, Rajeshwar P. Sinha

**Affiliations:** 1Laboratory of Photobiology and Molecular Microbiology, Centre of Advanced Study in Botany, Institute of Science, Banaras Hindu University, Varanasi 221005, India; kumarivarsh931@gmail.com (V.K.S.); jha422607@gmail.com (S.J.); ranapalak271@gmail.com (P.R.); mishrasona227@gmail.com (S.M.); nehayadavbhu123@gmail.com (N.K.); 2Taurmed Technologies Pvt Ltd., 304, Pearl’s Business Park, Netaji Subhash Place, New Delhi 110034, India; dr.sureshchandrasingh@gmail.com (S.C.S.); shekharxanand@gmail.com (S.A.); 3Department of Microbiology, Parul Institute of Applied Science, Center of Research for Development, Parul University, Vadodara 391760, India; drvijaysemilo@gmail.com; 4University Center for Research & Development (UCRD), Chandigarh University, Chandigarh 140413, India

**Keywords:** mycosporine-like amino acids, photoprotection, photo repair, resilience, scytonemin, ultraviolet radiation

## Abstract

Ultraviolet radiation (UVR) tends to damage key cellular machinery. Cells may adapt by developing several defence mechanisms as a response to such damage; otherwise, their destiny is cell death. Since cyanobacteria are primary biotic components and also important biomass producers, any drastic effects caused by UVR may imbalance the entire ecosystem. Cyanobacteria are exposed to UVR in their natural habitats. This exposure can cause oxidative stress which affects cellular morphology and vital processes such as cell growth and differentiation, pigmentation, photosynthesis, nitrogen metabolism, and enzyme activity, as well as alterations in the native structure of biomolecules such as proteins and DNA. The high resilience and several mitigation strategies adopted by a cyanobacterial community in the face of UV stress are attributed to the activation of several photo/dark repair mechanisms, avoidance, scavenging, screening, antioxidant systems, and the biosynthesis of UV photoprotectants, such as mycosporine-like amino acids (MAAs), scytonemin (Scy), carotenoids, and polyamines. This knowledge can be used to develop new strategies for protecting other organisms from the harmful effects of UVR. The review critically reports the latest updates on various resilience and defence mechanisms employed by cyanobacteria to withstand UV-stressed environments. In addition, recent developments in the field of the molecular biology of UV-absorbing compounds such as mycosporine-like amino acids and scytonemin and the possible role of programmed cell death, signal perception, and transduction under UVR stress are discussed.

## 1. Introduction

Cyanobacteria are a phylogenetically primitive group of Gram-negative photosynthetic prokaryotes with a wide distribution ranging from hot springs to the Arctic and Antarctic regions. These oxygen-evolving organisms appeared during the Precambrian era (between 2.8 and 3.5 × 10^9^ years ago) and provided favourable conditions for the evolution of current aerobic life [1]. Several natural compounds with therapeutic, commercial, and agricultural value are derived from cyanobacteria. They are also used as traditional energy resources and as an alternate source of natural compounds for cosmetics. Some cyanobacterial species are used as non-traditional sources of protein and food [2].

Ultraviolet radiation is divided into three categories: UV-C (100–280 nm), UV-B (280–315 nm), and UV-A (315–400 nm). While UV-A and UV-B rays are transmitted through the atmosphere, all UV-C and some UV-B rays are retained by the ozone layer. UV-C radiation never reaches the Earth’s surface. The continuous release of anthropogenic pollutants such as organobromides (OBs) and chlorofluorocarbons (CFCs) has led to ozone layer depletion. This results in an increased incidence of ultraviolet radiation (UVR; 280–400 nm) on the Earth’s surface, which is absorbed by biomolecules such as proteins and nucleic acids, ultimately resulting in lethal effects on biological systems [3]. Both photosynthesis and nitrogen fixation are energy-dependent processes driven by solar energy. The harvesting of solar radiation exposes cyanobacteria to harmful doses of UV-B and UV-A radiation simultaneously in their natural habitats. 

High exposure to UV-B radiation negatively impacts the production of food, ecological systems, and human well-being [4,5]. High-energy UV-B radiation has the greatest potential for cell damage caused by both its direct effects on DNA and proteins and its indirect effects via the production of reactive oxygen species (ROS) [6,7,8]. There are several targets for these potentially toxic ROS, including lipids, DNA, and proteins. Moreover, damage to the photosynthetic apparatus is also partially mediated by ROS, resulting in the inhibition of photosynthesis [1]. In contrast, UV-A radiation that is not absorbed directly by DNA can still induce DNA damage either by producing a secondary photoreaction of existing DNA photoproducts or via indirect photosensitizing reactions [8]. UVR has been reported to affect DNA and its morphological characteristics as well as cells’ differentiation, elastic properties, phycobiliprotein composition, protein profile, development, survival, pigmentation, orientation, metabolic processes, and ^14^CO_2_ uptake [9,10].

In response to the devastating effects of UVR, cyanobacteria have evolved a number of defence strategies, including migration and mat formation, efficient DNA repair mechanisms, including photoreactivation, excision repair, the SOS response, the production of antioxidants, the biosynthesis of UV-absorbing compounds such as MAAs and scytonemin, and apoptosis (or programmed cell death, PCD) [2]. However, little information is available regarding the molecular mechanisms of UV-absorbing compounds, the detection of UV signals, and UV-induced PCD in cyanobacteria. The purpose of this review is to collect, evaluate, and assess different studies related to the mitigation strategies of cyanobacteria under UVR stress and determine the critical topics related to this field. This review reports current updates in the field of the tolerance mechanisms, morphology, physiology, biochemistry, and molecular mechanisms of cyanobacteria under UVR stress. UV-mediated signal transduction and UV-induced PCD in cyanobacteria are also critically addressed.

## 2. Impact of UVR on Cyanobacteria

The surface of the Earth receives very small amounts of solar UVR (UV-C, 0%; UV-B, <1%; UV-A, <7%), but this part of the solar spectrum is extremely energetically active [1]. In cyanobacteria, there are several direct targets for harmful UV-B radiation, such as proteins and DNA, which have absorption maxima in this region, whereas UV-A irradiation has indirect effects through energy transfer from UV-A-stimulated chromophores to the DNA target [1]. After being exposed to UV-B radiation for 9 h, the levels of photosynthetic pigments, total chlorophyll, total carotenoids, and c-phycocyanin were reduced in *Arthrospira platensis* [11]. According to Vega et al. [12], numerous cyanobacteria and microalgae are affected by UV-B exposure in terms of their development, survival, pigmentation, orientation, growth, general metabolism, photosynthesis, nitrogen fixation, and nitrogen uptake (Figure 1).

### 2.1. Photosynthesis

UVR inhibits a number of photosynthetic processes in cyanobacteria, including the uptake of ^14^CO_2_, O_2_ evolution, and ribulose-1,5 bisphosphate carboxylase (RuBISCO) activity. RuBISCO, a holoenzyme, consists of two subunits: the 55 kDa larger subunit (LSU) and the 14 kDa smaller subunit (SSU) [13]. When exposed to UV-B radiation, RuBISCO becomes susceptible to a variety of modifications, including photo-degradation, polypeptide chain fragmentation, denaturation, active site modification, and the enhanced solubility of membrane proteins [14]. The availability of ATP and NADPH_2_ may also be reduced, which could prevent the fixation of carbon dioxide. Additionally, UV-B radiation negatively impacts tyrosine electron donors, quinine electron acceptors, and D1 and D2 proteins of photosystem II (PSII) [15]. Damage to the water-oxidizing Mn cluster in the PSII reaction centre (RC) leads to electron transport chain (ETC) inactivation [16]. The cyanobacterium *Anabaena variabilis* PCC 7937, when exposed to UVR, showed a reduction in the overall photosynthetic yield because of a reduction in the relative electron transport rate (rETR) [17]. Under UV-B radiation, *Spirulina platensis* showed a distorted thylakoid membrane with decreased Chl *a* content [18]. After exposing a *Phormidium* strain to UV radiation for only a few minutes, Häder [19] observed a reduction in O_2_ evolution. Additionally, UV-B radiation causes oxidative damage which leads to the lipid peroxidation of polyunsaturated fatty acids (PUFAs), which in turn weaken the strength of cells and thylakoid membranes and harm photosynthetic components [20].

### 2.2. Growth, Cell Differentiation, and Motility

UV-B irradiation severely affects cyanobacteria’s biochemical and physiological life processes, such as morphology, survival, cell differentiation, growth, development, pigmentation, orientation, and motility [21]. Döhler et al. [22], Häder et al. [23], and Newton et al. [24] reported the inhibitory effects of UV-B radiation on certain cyanobacteria, such as *Anabaena flos-aquae*, *Synechococcus leopoliensis*, and *Phormidium uncinatum*. It has been suggested that UV-B radiation damages cellular components that absorb radiation between 280 and 320 nm, leading to cell death. The tolerance of different species to UV-B radiation varies, and even strains that are closely related exhibit differences in sensitivity. In Antarctica cyanobacterium, *Oscillatoria priestleyi* growth was completely suppressed, whereas it was 62% in the case of *Phormidium murrayi* following a similar dosage of UV exposure. UVR significantly reduces the proportion of motile filaments and reduces the linear velocity of cyanobacterial cells, which affects their capacity to protect themselves from harmful UVR [2]. After 10–30 min of UVR exposure, a study has shown a significant reduction in the number of motile filaments of *Anabaena variabilis, Oscillatoria tenuis*, and *Phormidium uncinatum*. UV-B radiation hinders the ability of cyanobacteria to establish themselves in their photo environment, which ultimately leads to their premature death [25]. The differentiation of vegetative cells into heterocysts or akinetes and the fragmentation of filaments have been reported in the cyanobacterium *Anabaena siamensis* TISTR-8012 due to exposure to UV-B radiation [2]. Under UV-B radiation, the cyanobacterium *Anabaena* sp. PCC 7120 showed heterocyst differentiation of vegetative cells and a reduction of up to 49% in trichome length [9]. The three main heterocyst polypeptides (26, 54, and 55 kDa) are believed to be depleted after UV-B treatment as a result of the breakdown of the multi-layered heterocyst wall, which is crucial for maintaining the active form of the enzyme nitrogenase [26].

### 2.3. Nitrogen Metabolism

Nitrogenase is the key enzyme for nitrogen fixation, and UV-B radiation also significantly inhibits the process of nitrogen fixation, either directly or indirectly, due to the highly sensitive nature of the nitrogenase enzyme to UVR [27]. The activity of nitrogenase in a *Nostoc* species was lost after 45 min of exposure to UV-B, but nitrate reductase and the glutamine synthetase activity were found to be unaffected [28]. Strong protection against nitrogenase inactivation was provided by ascorbic acid or reduced glutathione. By preserving the sulfhydryl groups or disulfide bonds, reducing agents are known to maintain the native structure of various proteins in their original condition. Nitrogenase is extremely sensitive to oxygen and is likely to exhibit changes in its sulfhydryl groups. In different species, different levels of UV-B protection mechanisms have been observed, which may account for the variation in the amount of time needed to completely kill and inactivate nitrogenase activity [29]. Many researchers believe that the high susceptibility of nitrogenase to UV-B radiation is caused by either the presence of aromatic amino acids or the enzyme’s native structure [25].

### 2.4. Biomolecules

UV radiation has been shown to have an effect on the structural and functional integrity of accessory photo-harvesting pigments, such as phycocyanin, phycoerythrin, and allophycocyanin, in the marine cyanobacterium *Lyngbya* sp. A09DM [30]. UV-B radiation severely affects low-molecular-weight proteins. In *Nostoc* sp., αβ monomers of phycocyanin with a very low size (approximately 20 kDa) were the most affected. When *Nostoc carmium* and *Anabaena* sp. were exposed to UV-B radiation for 90 or 120 min, proteins with a size of 14.5–45 kDa were completely lost, but proteins with a size of about 55–66 kDa were unaffected, even after 120 min of UV-B irradiation. After 150 min of exposure to UV-B radiation, the proteins in *Nostoc commune* and *Scytonema* sp. completely disappeared [31]. The total protein profile of *Nostoc spongiaeforme* and *Phormidium corium* was changed both qualitatively and quantitatively after UV-B radiation and high light treatment [32]. The number and quantity of protein bands in various cyanobacteria were found to decrease linearly with the increased duration of UV exposure.

When cyanobacteria are directly exposed to UVR, their DNA is subjected to several kinds of damage and their cells also suffer from oxidative stress. One such type of damage that is specifically caused by UV-B radiation is covalent linkage between the bases, which results in lesions such as cyclobutane pyrimidine dimers (CPDs), pyrimidine photoproducts (6-4PPs), and their Dewar isomers. These CPDs and 6-4PPs DNA lesions produced by UV-B radiation can produce primary and secondary breaks in DNA, respectively [33]. These breaks stall the DNA polymerase and prevent it from progressing, thus inhibiting transcription and translation machinery [33], which eventually results in mutations or the death of the organism [34]. Numerous filamentous and unicellular cyanobacterial species such as *Anabaena* sp., *Nostoc* sp., and *Scytonema* sp. produce thymine dimers (T< >T) when exposed to UV radiation [35]. Thymine dimers appear more frequently under continuous UVR exposure. A cyanobacterium, *Arthrospira platensis*, formed CPD under UV stress in a way that was dependent on the temperature and its biomass. UV-induced DNA damage is reported in *Anabaena variabilis* PCC7937 and *Synechocystis* PCC 6308. Mosca et al. [36] observed an increased accumulation of DNA lesions in dried UV-irradiated biofilms compared to dried biofilms of the desert cyanobacterium *Chroococcidiopsis*.

## 3. UV-Mediated Signal Transduction in Cyanobacteria

To combat the negative effects of UVR, cyanobacteria have adopted various mitigation and defensive strategies. To adapt to changing conditions, cells must first recognize environmental cues and then transmit those signals to the proper machinery to mediate the responses. UV-B photoreceptors and a number of distinct signaling pathways used to detect low-level UV-B signals are found in higher plants. Secondary messengers, such as calcium, kinases, and the catalytic production of ROS, are examples of these pathways. From ultraviolet to far-red light, cyanobacteria respond to a variety of light sources. They possess some light-perceiving systems, including phytochromes, UV-A/blue photosensors, and the as-of-yet undefinable photoreception systems of UV-B-mediating responses [37]. These extremely specialized photoreceptors enable cyanobacteria to detect light direction, spectral ranges, and intensity. Cyanobacteriochromes (CBCRs) are linear tetrapyrrole-binding phytochrome-related photoreceptors. In contrast to the red/far-red reversible photoconversion of plant phytochromes, CBCRs display distinctive and varied photochemical characteristics as they can sense varied colours of light [38]. The chromophore-binding regions of plant phytochromes consist of a cGMP-phosphodiesterase/adenylate cyclase/FhlA (GAF) domain along with two other domains, while CBCRs consist of a single GAF domain. Green and red, UV and blue, or unidirectional photoconversion from violet to yellow absorption states have all been seen in the photoactive GAF domains of CBCRs [39,40]. Numerous phytochrome-related CBCRs have been connected to physiological processes that depend on light, including chromatic adaptation and phototaxis [39].

Instead of using light to control a behavioural cue, essential signaling pathways regulate movement towards or away from weak or strong light. Light waves with wavelengths ranging from the green (560 nm) to the red (720 nm) area mediate positive phototaxis in *Synechocystis* while UV-A light and blue (470 nm) light mediate negative phototaxis [41]. The three classes of photosensors reported in *Synechocystis* sp. PCC6803 that regulate UV-A-induced phototaxis are CRY-related cyanopterin Sll1629 [42,43]; CBCRs including PixJ1 (or TaxD1) [44], Cph2 [45], and BLUF photosensor PixD; and the ETR1-related UV intensity sensor (UirS) [46] (Figure 2a).

UirS, a negative UV phototaxis sensor in *Synechocystis* encoded by the *slr*1212 locus, has a histidine kinase (HiK) domain at the C-terminal end and transmembrane (TM) helices at the N-terminal end. With two conserved *Cys* residues in its phycocyanobilin (PCB)-binding GAF domain, UirS is a member of the CBCR subfamily. The ethylene binding in UirS is carried out by its TM helices, which are similar to the ethylene-binding domain of the *Arabidopsis* ethylene receptor (ETR1) [45]. However, its GAF domain can bind to the linear tetrapyrrole PCB to produce a CBCR that can switch between violet and green light [46]. One of *Synechocystis’* 16 two-component signaling gene clusters encodes UirS [47]. The genes *slr*1213 and *slr*1214 encode UirR (UV intensity response regulator) and LsiR (light and stress integrating response regulator), respectively. The C-terminal end of UirR consists of the DNA-binding domain, and a receiver domain is present at the N-terminal end. UirS-UirR is a two-component signal transduction system which targets the response regulator LsiR. LsiR comprises a UV-specific photosensory signaling system. In order to maximize its absorption in the UV-violet (Puv) and UV-green (Pg) spectral bands, UirS-GAF has two photointerconvertible forms. After Puv is exposed to UV-A, Pg is produced. Pg has two minor peaks in the UV area (325 nm and 382 nm) and absorbs most of the light maximally in the green region (534 nm). Green light was found to revert the Pg spectrum to the Puv spectrum, with maximum absorption at 382 nm [46]. While LsiR is a response regulator of the PatA subtype, UirR serves as the transcription activator of the stress response. Inorganic carbon and high levels of H_2_O_2_ have been shown to cause the induction of PatA-type response regulators in *Synechocystis* [48,49]. The membrane-bound two-component signaling system UirS-UirR detects high UV-A fluence rates and starts the signal transduction cascade primarily by phosphorylating UirR, which then triggers the transcription of the response regulator LsiR. As a result, LsiR acts as a regulator of the signal output to control the negative UV-A phototaxis signaling pathway (Figure 2b). According to Apel and Hirt [50] and Hancock et al. [51], increased intracellular hydrogen peroxide (H_2_O_2_) may be necessary for UirR activation via phosphorylation at high UV-A fluence rates. The UirS-UirR two-component signaling system present in *Synechocystis* performs a dual function as it can simultaneously perceive high-intensity UV-A while also serving as a target for ROS. This dual function is found to mediate downstream signaling to initiate UV-induced negative phototaxis [45].

Current knowledge of the underlying molecular mechanisms of UV-B signal transduction in cyanobacteria is limited. Additionally, cyanobacteria exposed to UV-B light produce ROS, which may have a role in signal conversion under UV stress. According to Richter et al. [52], ROS in *Arabidopsis* raise the intracellular calcium concentration, which, along with calmodulin, controls the catalase activity. *Anabaena* sp. and *N. commune* both possess UVR-sensitive L-type calcium channels [1]. Richter et al. [53] discovered that the *Anabaena* sp. increases its intracellular calcium concentration in response to UV-B exposure. This suggests that cyanobacteria utilize calcium as well as calmodulin-like proteins to translate UV signals. An essential part of calcium-dependent signaling has also been identified in the cyanobacterium *Nostoc* sp. [1]. Methyl viologen also caused UV-shock proteins to be expressed in *Anacystis nidulans* R-2, demonstrating the role of ROS in transducing UV signals. Consequently, it is probable that during UVR exposure, cyanobacteria may control a number of adaption mechanisms by transducing UV-B signals via ROS and calcium. Cyanobacteria also contain other signaling molecules, including cAMP and cGMP, and it has been discovered that changes in cGMP’s cellular homeostasis play a role in signal transduction during UV exposure [1].

## 4. UV Stress Tolerance and Mitigation Strategies in Cyanobacteria

UVR can affect many parts of cyanobacteria’s cells, either directly or indirectly. These bacteria have found several ways to protect themselves from this radiation, which lets them grow and live in places with high light intensities (Figure 3).

### 4.1. Avoidance and Migration

Several cyanobacteria have learned to avoid areas with high light intensities as a first line of defence. Transitioning from higher to lower amounts of solar radiation is one of the several ways to avoid solar radiation. Other ways include mat formation, changes in shape to improve self-shading, and the production of extracellular polysaccharides. In order to avoid intensive solar radiation, mobile cyanobacteria can migrate downward into mat communities or go down into water columns [54]. According to Wu et al. [31], *Arthrospira platensis* has an effective defence mechanism against photoinhibition, and spirals of *A. platensis* become more compressed when exposed to UV-B radiation, which ultimately leads to self-shading. Ehling-Schulz et al. [55] hypothesized that UV resistance is conferred by extracellular glycan synthesis in *Nostoc commune* by expanding the radiation’s effective route. In response to UVR, ROS such as superoxide (O^2−^), hydroxyl (OH^−^), and hydrogen peroxide (H_2_O_2_^−^) radicals are produced in amalgamation with oxygen and other organic substances, resulting in oxidative stress [45]. It has been reported that *Oscillatoria* sp. and *Spirulina* cf. *subsalsa* migrate vertically every day to evade periods of intense sun irradiation [56]. When exposed to low PAR (20–90 Wm^−1^) or complete darkness, mobile *Spirulina* and *Oscillatoria* exhibit upward movement in the hypersaline mats of Guerrero Negro. Compared to UV-A and PAR, *Microcoleus chthonoplastes* found in Solar Lake, located in Sinai, Egypt, drift more commonly in UV-B radiation, signifying that UV-B is the most active wavelength [57]. By migrating and moving vertically within mats or sediments, cyanobacteria can conserve energy for photoprotective adaptation [58].

### 4.2. Mat Formation

Cyanobacteria form mat-like or crust-like formations that range in thickness from a few micrometres to a few decimetres and are firmly attached to the substrate. Ten to forty species of distinct cyanobacterial taxa frequently form colonies [59]. The composition of cyanobacterial mats varies according to environmental factors, substrate types, and the capabilities and characteristics of colonizing species. Massive filamentous cyanobacteria often colonize first, likely due to their extensive extracellular sheaths or mucus layers, increasing their water retention properties [60]. *Lyngbya aestuarii* and *Calothrix* sp. predominate in marine environments, such as the hypersaline Laguna Guerrero Negro in Mexico [61]. In addition, Karsten et al. [62] noted that a mat connected to a subtropical mangrove system that was developing in direct sunlight contained two distinct cyanobacterial layers. *L. aestuarii*, which was present in the upper layer, contained scytonemin, whereas *Microcoleus chthonoplastes* lacked this photoprotective compound. *Lyngbya*’s scytonemin provided a photoprotective mechanism for *M. chthonoplastes* [62].

In the ponds of industrial salterns in Eilat, Israel, gypsum crusts have been reported [63]. Crystals of gypsum are resistant to UVR, abrupt temperature changes, and dehydration. In the crusts, two distinct horizontal oxygenic phototrophic communities were exposed to vastly different radiation levels. The layer with *Halothece*-like cyanobacteria containing carotenoids produced the most oxygen. The rate of oxygen production in this stratum responded significantly to the radiation levels. Due to the presence of scytonemin or gloeocapsin, terrestrial substrates frequently have black or dark-coloured cyanobacterial crusts [64]. Species in severe ecosystems develop extremely thick outer sheath layers to withstand the intense solar radiation and arid conditions, particularly during the summer. The most common species of aeroterrestrial cyanobacteria are *Nostoc*, *Scytonema*, *Calothrix*, and *Tolypothrix* species [65].

### 4.3. Antioxidant Systems

Cyanobacteria have devised an antioxidant system to mitigate oxidative stress as a secondary defence against UVR. This system includes both non-enzymatic and enzymatic antioxidants. The non-enzymatic antioxidants are carotenoids, α-tocopherol (vitamin E), ascorbate (vitamin C), and reduced glutathione. Superoxide dismutase (SOD), catalase, glutathione peroxidase (GSH-Px), ascorbate peroxidase (APX), dehydroascorbate reductase (DHAR), glutathione reductase (GR), and monodehydroascorbate reductase (MDHAR) are the enzymatic antioxidants [20]. Carotenoids capture triplet-state energy produced from chlorophyll and safeguard cells from photooxidation-related harm while lipid peroxidation is averted by α-tocopherol via foraging ROS [66]. In the tropical Guyana-shield region of Venezuela, terrestrial cyanobacteria had the highest canthaxanthin/carotene concentrations and the highest carotenoid/chlorophyll ratio in response to UV-A and UV-B radiation, respectively, indicating the photoprotective function of these pigments [67,68]. *Microcystis aeruginosa* increased carotenoid production to combat UV-B-induced ROS production [69]. Carotenoids in *Synechococcus* PCC 7942 function as antioxidants by neutralizing UVB-induced radicals in the photosynthetic membrane [70]. In addition, it has been discovered that carotenoids attached to the outer membrane provide a rapid active SOS response to stop impending cell harm [67].

Ascorbate directly neutralizes ROS, restores α-tocopherol, and functions as a substrate for APX and violaxanthin de-epoxidase processes. Ascorbate and α-tocopherol are regenerated by the glutathione-ascorbate cycle of α-tocopherol and ascorbate. Additionally, the thiol groups present in several enzymes are protected [71]. SOD is the only antioxidant enzyme that scavenges the superoxide anion by converting this free radical to oxygen and hydrogen peroxide. The carotenoid motif in *Nostoc commune* is modified in the presence of UV-B irradiation, acting as an outer-membrane-bound UV-B shield [67]. In response to UVR, SOD and APX activity increased in *N. spongiaeforme* and *Phormidium corium* [16]. *Anabaena* sp. had a greater survival rate because ascorbic acid and N-acetylcysteine (NAC) diminishe UVR-induced DNA strand disintegration and lipid peroxidation [72]. *N. commune* found in a dehydrated field showed an increase in the content of active iron superoxide dismutase (FeSOD) to reduce the damaging effects of oxidative stress that were result of constant cycles of dehydration and rehydration under in situ UV-A or UV-B irradiation [73].

### 4.4. UV-Screening Compounds and Their Molecular Mechanisms

Cyanobacteria synthesize UV-absorbing molecules, such as MAAs and scytonemin, as a defence against continued solar radiation [74]. MAAs and scytonemin are eminent UV-shielding molecules that protect organisms against harmful UVR [75,76]. There is no specific photoreceptor for various UV rays reaching on Earth, but an indirect signal perception has been observed in case of UV-B irradiation [77]. A novel UV-B photoreceptor found in *Chlorogloeopsis* PCC6912 induces mycosporine-like amino acid (MAA) production in this cyanobacterium. MAAs are water-soluble, colourless compounds with a size smaller than 400 Da, consisting of an aminocyclohexenone or an aminocyclohexenimine ring with aminoalcohol or nitrogen substituents [78]. The varied absorption spectra of various MAAs are influenced by different side groups attached to the central ring structure and nitrogen substituents. The UV absorption maxima of MAAs (309–362 nm) cover the range of UVR reaching Earth. It has high molar extinction coefficients (ε = 28,100–50,000 M^−1^ cm^−1^), which show the ability of MAAs to absorb at specific wavelengths. Commonly synthesized MAAs in different cyanobacteria are listed in Table 1. These properties along with their ability to prevent the formation of reactive oxygen species (ROS), antioxidant properties, and resistance to a variety of abiotic stressors such as temperature, UVRs, several solvents, and pH validates the role of MAAs as photoprotective compounds [79]. The primary function of MAAs is to act as a UV-screening compound, which can be conferred by the direct relation between its induction and the exposure of an organism to UVRs [80]. By absorbing highly powerful UVR and then releasing this energy as harmless heat radiation into their environment, MAAs shield cells from UVR [81].

MAA induction is triggered indirectly by the ROS species produced due to the oxidative stress created by PAR, UV-A, and UV-B radiation, with UV-B being the most potent inducer because it possesses the highest energy [83]. If the ROS produced by oxidative stress are not removed promptly, they can result in DNA damage, mutations, and a reduction in DNA repair. Antioxidants can help avoid oxidative stress by reducing the damaging effects of ROS. MAAs not only absorb and dissipate the excess heat generated by UVR, but they can also act as free radical scavengers due to their antioxidant activity [84]. Once the MAAs’ induction is triggered, it quenches the ROS species, ROS species react with it, and the hydrogen atom from the C-4 or C-6 carbon is abstracted by the free radical, resulting in the delocalization of the radical electron over the double bonds of the cyclohexane ring, thus conferring stability to the free radical, e.g., the 2,20-azo-bis (2 amidinopropane), dihydrochloride (AAPH), and 2,2′-azinobis (3-ethylbenzothiazoline-6-sulfonic acid) (ABTS) radicals. The scavenging activities of MAAs depend on the cyclic ring structure associated with them. In contrast to imine-type MAAs (aminocyclohexenimine ring structures), carbonyl-type MAAs (aminocyclohexenone ring structures) have a significantly higher radical-scavenging activity at physiological pH. The imine type of MAAs shows changes in radical scavenging activity in a pH-dependent manner, the decreased delocalization of electrons at a low pH, and normal delocalization in basic surroundings (Figure 4). Thus, MAAs containing an aminocyclohexenone structure play a key role in the antioxidant mechanism of the MAAs [85]. Consequently, MAAs have antioxidant properties, as well as the ability to suppress lipid peroxidation, AAPH, 2,2-diphenyl-1-picrylhydrazyl (DPPH) and ABTS radical extinction, singlet oxygen quenching, superoxide anion radical scavenging, hydrogen peroxide extinction, and physiological activities [86,87]. MAAs typically reside in the cytoplasm of the cell, except for in *Nostoc commune* in which they accumulate in the extracellular space, providing more effective protection against UVR [67]. The recently discovered MAAs from *Klebsormidium*, klebsormidin A, and klebsormidin B, have demonstrated that UVR exposure dramatically induces their production and intracellular enrichment, indicating the role of these molecules as natural UV sunscreens [88].

Scytonemin, a yellow/brown, lipid-soluble pigment, found in the extracellular sheath, a dimer of indolic and phenolic subunits with a molecular mass of 544 Da, is a UV-screening pigment of the alkaloid class that is found exclusively in cyanobacteria [10]. Two subunits (indolic and phenolic subunits) of the scytonemin dimer are linked to each other by an olefinic carbon atom. Scytonemin is capable of absorbing all wavelength of UVR, including UV-A, UV-B, and UV-C radiation, specifically at 252 ± 2, 278 ± 2, and 300 ± 2 nm, but its maximum absorption is observed at 386 ± 2 nm [89]. Scytonemin exists in two forms: oxidized (yellow) and reduced (red), which are known as fuscochlorin and fuscorhodin, respectively. Various syctonemin obtained from different cyanobacteria are dimethoxyscytonemin, tetramethoxyscytonemin, scytonin, and scytonemin-3a-imine [10]. The biosynthesis of scytonemin is induced mostly by UV-A radiation. High temperatures/oxidative stress along with UV-A radition and restricted nitrogen accessibility have been observed to trigger the synthesis of scytonemin in the *Chroococcidiopsis* sp./*Scytonema* sp. R77DM and *Nostoc punctiforme* PCC 73102, respectively [90]. The molar extinction coefficient (*ε*) of scytonemin is 250 g^−1^ cm^−1^ at a wavelength of 384 nm, indicating its potency as an effective UV screening compound that protects cyanobacteria from harmful UV-A/B radiation entering the atmosphere, allowing them to thrive. Its oxidized form is more prevalent and exhibits less solubility in tetrahydrofuran, pyridine, and organic solvents, whereas the reduced form shows more solubility in organic solvents [91]. The production of scytonemin was influenced by UV-A radiation, while blue, green, and red light do not affect its synthesis. Scytonemin has been found to reduce the amount of UV-A radiation penetrating cyanobacterial cells by 90% [92]. Even during desiccation, scytonemin is enormously stable and can perform its screening function without incurring the additional metabolic expenditure [93]. Along with its UV-screening activity, scytonemin shows strong antioxidant activity and slow radical scavenging activity. It scavenges radicals such as DPPH or ABTS [94]. Its antioxidant activity was shown to be dose dependent, and when using 0.5 mM ascorbic acid as a positive control, 22% and 52% activities were found at concentrations of 0.4 and 0.8 mM, respectively. The stability of scytonemin was investigated under different abiotic stress conditions, such as heat (60 °C), UV-B (0.78 W m^−2^) treatment, and H_2_O_2_ (0.25%). Scytonemin was only partially degraded in the presence of heat and the UV-B treatment as compared to 0.25% H_2_O_2_ stress, for which more prominent degradation was observed [92]. Thus, both MAAs and scytonemin are the crucial components of the mitigation strategy implemented by cyanobacteria against UVR falling on these organisms in extreme heat conditions, allowing them to survive intense heat and providing them with a wide distribution.

### 4.5. Repair and Resynthesis

When UVR bypasses the first, second, and third lines of protection, the repair and resynthesis of complex targets in cyanobacteria become crucial as a fourth line of defence. Multiple DNA repair mechanisms exist, which make organisms more resistant to radiation [95]. These include direct or excision repair, recombinational repair, and light-dependent photoreactivation by photolyase. “Photoreactivation” is the most dominant repair mechanism in cyanobacteria [33], which removes CPDs or 6-4PPs dimers formed due to exposure of these organisms to UV radiation. Photolyase enzymes such as CPD photolyase and 6-4 photolyase utilize UV/blue light for this kind of repair. Photolyase has two cofactors, a catalytic cofactor, such as 5,10-methenyltetrahydrofolate (MTHF), 8-hydroxy-5-deaza-riboflavin (8-HDF), or flavin mononucleotide, and a light-harvesting cofactor, such as deprotonated reduced flavin adenine dinucleotide (FADH). MTHF or 8-HDF (λ_max_ ~380 and ~450 nm, respectively) absorb long-wavelength UV-A or blue light energy and are transferred to the catalytic cofactor FADH. FADH in its excited state transfers the electrons to the CPD dimer, hence resulting in the monomerization of these dimers [2]. *Anabaena* sp. PCC 7120, *A. variabilis* PCC 7937, *Anabaena* sp. M-131, and *A. variabilis* PCC 7118 have displayed the photoreactivation mechanism [96]. *Synechocystis* sp. PCC 6803 contains genes for photolyase homologs which have an important role in photoreactivation [97]. Other repair mechanisms such as direct or excision repair, recombinational repair, SOS repair, etc. are not exclusive to cyanobacteria; they operate in a manner similar to that of other organisms. For information on these DNA repair mechanisms, we refer to the review by Pathak et al. [21]. An enhanced recA transcript level was shown in *A. variabilis* as well as the profusion of the accompanying 37–38 kDa polypeptide in the presence of UVR [98]. In *Synechococcus elongatus*, the gene responsible for the DNA repair enzyme Fpg (formamydopyrimidine-DNA glycosylase) is expected to be involved in defence against oxidative damage [99]. Therefore, damages caused by UVR in cyanobacteria are repaired by excision repair and photoreactivation. Along with repairing damaged DNA, cyanobacteria are also able to replace fragmented copies of protein. Mosca et al. [36] performed a study on the anhydrobiotic *Chroococcidiopsis* sp. CCMEE 029 by drying the cyanobacterial film in air-dried storage and then rewetting it. On exposing this dried film to Mars-like UV radiation, they observed a marked over-expression of the *phrA* gene, suggesting the involvement of the codified photolyase in monomerizing CPD dimers. The *uvsE* gene, encoding for a putative UV damage endonuclease, was over-expressed during the recovery of dried biofilms, suggesting their involvement in DNA repair. Additionally, the dried UV-irradiated rewetted biofilms showed an increased expression of nucleotide excision repair genes of which *uvrC* showed the highest expression [36]. According to the study by Ehling-Schulz et al. [100], a distinct and intricate mechanism is present in *N. commune* that involves the use of 493 distinct proteins responsible for protection against stress originating from solar radiation.

## 5. UV-Mediated Cell Death/Apoptosis

Cyanobacteria can go through apoptosis or programmed cell death (PCD) when a cell is damaged in a way that cannot be fixed. *Trichodesmium* sp. and *Anabaena flos-aquae*, which fix nitrogen, have been shown to have a PCD that works independently. This PCD is caused by high irradiance, iron depletion, and oxidative stress [101]. *Microcystis aeruginosa*’s genome has recently been sequenced, and a number of genes encoding caspases, which are eukaryotic enzymes implicated in PCD, have been identified [102]. *Trichodesmium* sp. has caspase activity and proteins that react with human caspase-3 antibodies [101]. In plants and phytoplankton, ROS also play a role in PCD [103,104]. According to Lu et al. [105], ROS worked as a signal to boost caspase-3(-like) activity, which then initiated PCD in *M. aeruginosa* under pyrogallic acid stress. Another study similarly found that polybrominated diphenyl ethers in *Thalassiosira pseudonana* caused alterations in cellular ROS levels that were connected with PCD indicators and the dead cell ratio [106]. The increase in caspase-like activity occurred simultaneously with an increase in ROS in *M. aeruginosa* [107]. Thus, the ROS served as a signaling molecule that caused the test species’ caspase-3(-like) activity to increase (Figure 5). So, it is clear that PCD is essential for cyanobacteria that are under oxidative stress. However, more research is needed to determine whether this mechanism is beneficial or not under UV stress or oxidative stress caused by UVR.

## 6. Conclusions

An enormous amount of concern has been raised regarding the negative impacts of UVR on terrestrial and aquatic ecosystems due to the increased incidence of UVR in the Earth’s atmosphere as a result of the discharge of human ozone-depleting compounds. One of the most significant producers of biomass, cyanobacteria, is vulnerable to increased UVR stress, and several UV-induced effects have been reported in different cyanobacterial species. Through ROS-induced oxidative damage, UV-B radiation can directly or indirectly harm cellular DNA, proteins, and physiological and biochemical processes. Although ROS can control gene expression by serving as a second messenger in a number of cellular signaling pathways that trigger defence and resiliency mechanisms, its specific function in UVR stress signaling in cyanobacteria is still unclear. In cyanobacteria, great progress has been made in identifying the UV-A photoreceptor and a number of molecules involved in the early stages of UV-A signaling pathways. The detrimental impact of UVR on this dominating microbiota could throw entire ecosystems out of balance. Cyanobacteria have evolved protective mechanisms and cellular machinery throughout the course of evolution, enabling them to live and thrive in a variety of sun-exposed habitats with high solar UVR. Cyanobacteria have an intriguing ability to reduce UVR toxicity by employing UV-absorbing/screening substances, such as MAAs and scytonemin. Though research into the molecular biology and functions of MAAs and scytonemin is still in its early phases, some gene clusters implicated in the manufacture of these chemicals have recently been characterized. Cyanobacteria’s ability to tolerate stress is also influenced by the SOS response and apoptosis (PCD). Even though our knowledge of many different UV effects has grown over the past few decades, there are still many unanswered questions that preclude a complete understanding of the UV tolerance mechanisms of cyanobacteria. Overall, it appears that cyanobacteria have evolved into the most ecologically successful prokaryotes on Earth mainly because of their dynamic stress tolerance systems.

## Figures and Tables

**Figure 1 ijms-24-12381-f001:**
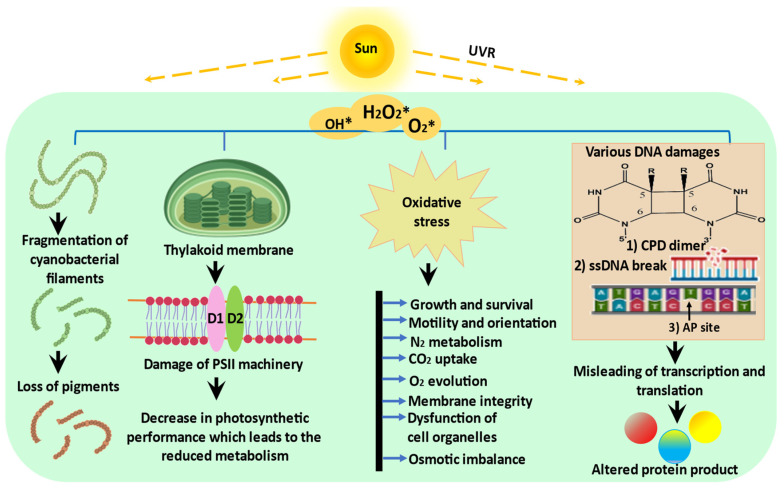
Schematic representation of possible effects of UVR on cyanobacteria: UVR exposure triggers the production of ROS which negatively affects the morphology, physiology, and biochemistry of cyanobacteria (modified from Rastogi et al. [2]).

**Figure 2 ijms-24-12381-f002:**
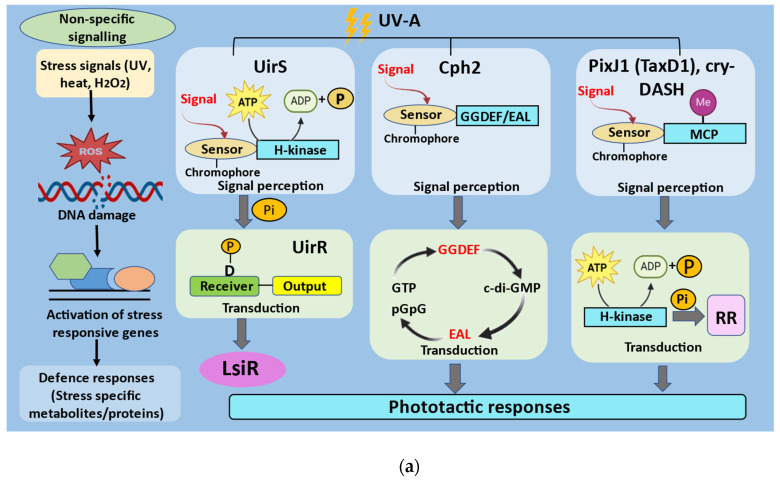
Schematic representation of UV-mediated signaling in cyanobacteria. The major processes involved in signaling cascade include signal perception, transduction, and cellular responses: (**a**) different classes of UV-A sensors, signaling molecules, and the UV-A signaling pathways in which they are involved. GGDEF and EAL, output domains; MCP, methyl-accepting chemotaxis protein signaling domain; RR, response regulators; (**b**) schematic representation of UirS/UirR/LsiR-based negative UV phototaxis signaling pathway in cyanobacteria. Two-component system initiates phosphorelay cascades, which consist of transfer of a phosphate moiety in the transduction of the light signal. Many of the targets of the phosphorelay are response regulator (RR)-type molecules (modified from Moon et al. [45] and Song et al. [46]).

**Figure 3 ijms-24-12381-f003:**
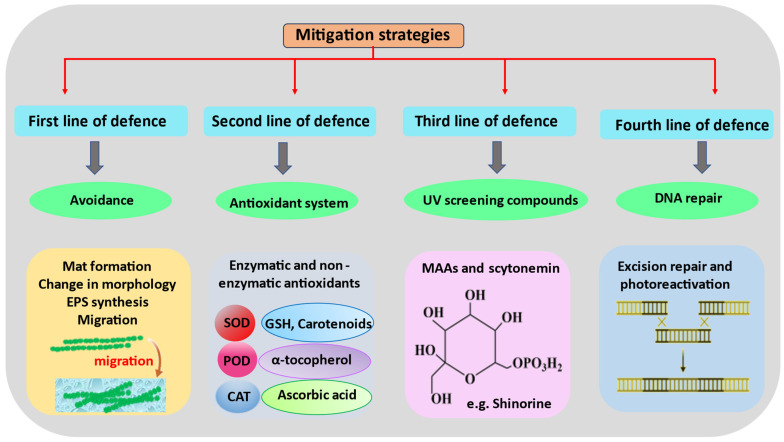
Schematic representation of various mitigation strategies adopted by cyanobacteria in order to counteract the detrimental effects of UVR.

**Figure 4 ijms-24-12381-f004:**
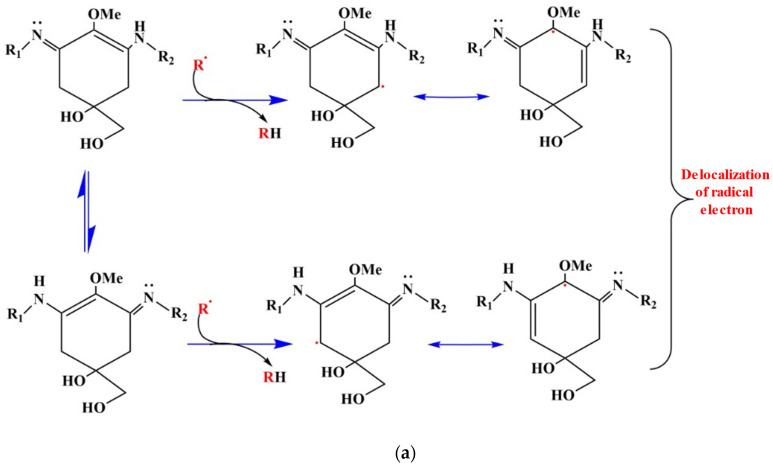
ROS scavenging mechanism of iminocyclohexene-type MAAs by resonance of radical electron at C4 and C6 position: (**a**) basic environment; (**b**) acidic environment (adapted from Wada et al. [85]).

**Figure 5 ijms-24-12381-f005:**
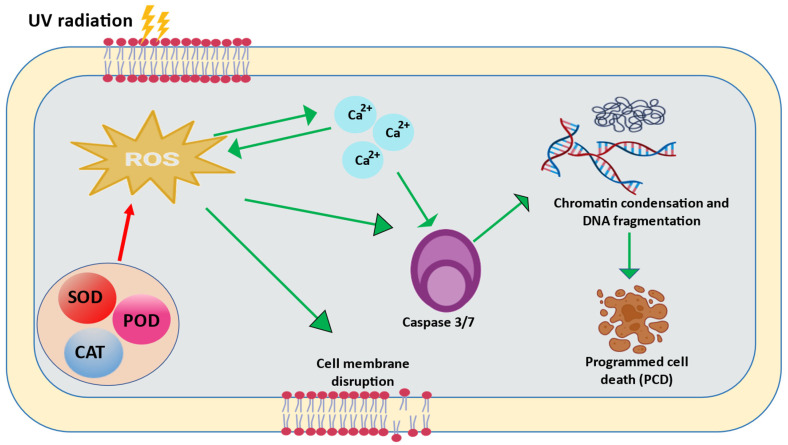
Proposed model for UV-induced PCD in cyanobacteria. UV-induced elevated levels of ROS promote the expression of caspase-like enzymes resulting in PCD. Red arrow signifies negative regulation while green arrows represent positive regulation (modified from Bai et al. [107]).

**Table 1 ijms-24-12381-t001:** An overview of cyanobacteria producing MAAs (adapted from Jain et al. [82]).

Order	Organism	MAAs (λ_max_ nm)
Synechococcales	*Synechocystis* sp. PCC 6803	Mycosporine-taurine (309); Dehydroxyl-usujirene (357)
Chroococcales	*Gloeocapsa* sp.	Shinorine (334); Mycosporine-glycine (310)
	*Aphanothece halophytica*	Mycosporine-2-glycine (334)
	*Euhalothece* sp.	Euhalothece (362); Mycosporine-2-glycine (334)
	*Microcystis aeruginosa*	Shinorine (334); Porphyra (334)
Oscillatoriales	*Lyngbya* sp. CU2555	Palythine (320); Asterina (330)
	*Microcoleus chthonoplastes*	Shinorine (334)
	*Oscillatoria spongelidae*	Mycosporine-glycine (310); Usujirene (357); Palythene (360)
	*Trichodesmium* sp.	Asterina (330); Shinorine (334); Porphyra (334); Palythene (360)
Nostocales	*Anabaena* sp.	Shinorine (334)
	*Anabaena doliolum*	Mycosporine-glycine (310); Porphyra (334); Shinorine (334)
	*Anabaena variabilis* PCC 7937	Shinorine (334); Palythine-Serine (320); MAA-glycine (310)
	*Nostoc commune*	Shinorine (334)
	*Scytonema* sp.	Shinorine (334); MAA-315 (315); Asterina (330)
	*Nostoc punctiforme* ATCC 29133	Shinorine (334)
	*Nostoc* sp. HKAR-2 and HKAR-6	Shinorine (334); Porphyra (334)
	*Nodularia* sp.	Shinorine (334); Porphyra (334)
	*Aphanizomenon flos-aquae*	Porphyra (334)
	*Chlorogloeopsis* PCC 6912	Mycosporine-glycine (310); Shinorine (334)

## Data Availability

Not applicable.

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
