# Peer review of "Resilience and Mitigation Strategies of Cyanobacteria under Ultraviolet Radiation Stress"

_ijms, 2023, doi:10.3390/ijms241512381_

Round 1

Reviewer 1 Report

This is a very extensive review dealing with cyanobacterial response to UV radiation. I would like to suggest improving the quality of Table 1 since the molecular structures look a little deformed.

Author Response

Reviewer 1

Comments and Suggestions for Authors

Issue 1: This is a very extensive review dealing with cyanobacterial response to UV radiation. I would like to suggest improving the quality of Table 1 since the molecular structures look a little deformed.

Response: We thank the reviewer for taking the time to assess our MS. We appreciate the reviewer’s insightful suggestion. We have provided the modified table summarizing the MAAs synthesis in different cyanobacteria. Please see page 10 of the revised MS, Table 1.

Reviewer 2 Report

The manuscript needs a lot of work. Please see attached comments.

Author Response

Reviewer 2

Comments and Suggestions for Authors

Issue 1: This manuscript is a review of cyanobacterial UV defence mechanisms, which is a subject with a very wide scope. As a result, the coverage is necessarily shallow, and encompasses discussion of mechanisms that are present in the majority of organisms (e.g. thymidine dimer formation, DNA repair mechanisms) and are not unique to cyanobacteria, without explaining to the reader that this is the case. Such coverage can be found in biochemistry textbooks and it is doubtful whether they should form the basis of a current review in a medium impact journal like IJMS. A focus on specific aspects of UV protection which are unique (or almost unique) to cyanobacteria would allow for more detail, however a number of recent reviews (e.g. “Unravelling photoprotection in microbial natural products” Woolley & Stavros 2019 as well as the author’s references [19;21;103]) already detail these areas, so the authors would need to show that their approach to the topic provided a perspective that was valuable despite the detailed coverage provided by these other reviews.

Response: We thank the reviewer for taking the time to assess our MS. We appreciate the reviewer’s insightful suggestion. We have discussed certain mechanisms like photoreactivation that are dominantly reported in cyanobacteria. Also, we have focussed on certain specific aspects and molecular mechanisms of UV-absorbing compounds and ROS-mediated cell death that are reported in cyanobacteria in response to UV radiation.

Issue 2: The writing style is amateurish, with bad grammar, many colloquialisms, redundant sentences, frequent sentences that provide generalisations that can’t be supported, which provide few specifics and no references. I started to enumerate the defects in the text but was soon overwhelmed. Therefore I do not provide a commentary detailing specifics.

Response: We thank the reviewer for pointing this out. We’ve corrected the grammatical errors to the best of our ability in the revised MS.

Issue 3: The manuscript is poorly organised, with repetition due to the fact that descriptions of defence strategies (e.g. motility) must be re-discussed in later sections (e.g. the discussion of signal transduction). There is little differentiation between UV defence strategies that are almost universal in biology (e.g. formation and repair of thymidine dimers), widespread in photosynthetic organisms (e.g. carotenoids) and those that are unique to cyanobacteria (e.g. MAAs and scytonemin). In several places, I questioned whether the authors really understood what they were discussing, although this could have been due to the poor standard of written expression (e.g. Rubisco is a holoenzyme with 8 large subunits, “…eight small subunits (SSU, 14 kDa), and tryptophans (Trps)”). The work of other authors is often reported without specifics so that the reader is none the wiser about what was demonstrated in the original article nor what its significance is. Table 1 contains poorly represented structures without a uniform appearance.

Response: We thank the reviewer for pointing this out. This observation is correct. We have re-organized and revised the text to address your concerns and hope that information conveyed in the revised MS is now clear. The subheadings (growth, survival and motility) of section 2 have been combined together within the revised MS. Please see page 3-4 of the revised MS, lines 95-104 and115-139. Also, we have provided the modified table summarizing the MAAs synthesis in different cyanobacteria. Please see page 10 of the revised MS, Table 1.

Issue 4: To be publishable, the manuscript would need to be narrowed in scope to cyanobacteria focussed mechanisms (with brief reference to widespread or universal UV defence mechanisms), considerably tightened, redundancy removed, reorganised to create a better flow, the written expression thoroughly revised, the many minor errors fixed, and the impact of the recent discoveries in this area delineated and the resulting further questions clearly described.

Response: We appreciate the reviewer’s insightful suggestions. We have removed the general discussions regarding UV defence mechanisms and focussed mainly on the works that are related to cyanobacteria. Also, we have re-organized the MS to create a better flow and make it clear.

Reviewer 3 Report

This paper is a review of the effects of UV radiation on cyanobacteria, focusing on their resilience and mitigation strategies. As a result, the review is a summary of recent findings about the processes used by cyanobacteria to tolerate high UV radiation. In general, the review is informative and precisely focused on a single subject. However, as detailed in the points below, I believe it lacks good organization. 

The abstract of the paper is sufficiently informative, but I believe it is missing one critical point, namely the need for this review article. I noticed some other reviews about UV radiation and cyanobacteria in the literature, so I asked the authors to explain why they decided to write this review, why this review is needed now, and what ultimately distinguishes this review from others. 

Although I am not a native English speaker, I recommend that the authors thoroughly check the manuscript for English correctness. Perhaps a professional English speaker would be beneficial. I found several grammatical errors, such as commas between the nouns and verbs within a sentence. 

What I previously mentioned for the abstract, namely the need for more in-depth justification for writing this review, also applies to the introduction. Around line 60, the authors simply stated that this review contains the most recent findings on cyanobacteria's response to UV stress, but more justification is required. 

The Introduction chapter is not well organized. The authors introduced the concept of programmed cell death at line 64, and they spent a few lines shortly afterwards describing the importance of PCD in the ecology of cyanobacteria. However, the authors introduced the concept of photoprotective compounds without any connection to PCD at line 70. Line 79 reiterated the scope of the review, stating that it is focused on the most recent findings, and so on. The review's scope should be presented only once. 

In all honesty, I don't see the point of figure 1, at least not in its current location. Figure 1 functions as a review summary because the top section describes the effect of UV radiation on cyanobacteria at various levels. The bottom section details the mitigation strategies. This figure, in my opinion, should be divided in two. While it makes sense to show the top part at the beginning of this review, the bottom part should be included in the second part. One thing to note about the top part of figure 1: it's a little confusing. I understand the authors' preference to report on all the effects of UV radiation on cyanobacteria. However, the picture is not easily understood. What does the graph represent? I believe the image should be restyled to make it simpler. 

The authors stated on line 106 that UVB radiation also affected that enzyme. What enzyme? 

The subchapter on growth and survival is too brief to warrant its own section. It might be better to combine this with the next one about mobility. Furthermore, the sentence that UV radiation harms the growth and survival of many cyanobacteria is overly broad. Is it present at all wavelengths? Is exposure time a factor? What about UVA and UVC exposure? 

I noticed that the authors used the broad term UVR at times and the more specific term UVB radiation at others. I believe that they should always specify what type of UV radiation has been studied. 

Changes in protein content are discussed in Subchapter 2.6. More information about specific proteins, in my opinion, is required. I'd like to check the article mentioned in reference 42, but the DOI link is broken, and I can't find it. 

What do the authors mean by "native DNA" on line 182? 

Figure 1b, on the other hand, I believe should be placed exactly where it is first mentioned. Although the figure is sufficiently clear, the purple box contains unreadable chemical structures. 

There are no bibliographic citations in chapter 3.1 about avoidance. 

Should subchapter 3.1 and 3.2 be combined? I'm wondering if avoidance is related to migration. 

I dislike table 1 because it is uninformative. Molecular structures are oversized in comparison to the table, and chemical structures differ in size as well. I believe that showing the molecular structures of UV-absorbing compounds should be useful if they are related to absorbance features; otherwise, they are just drawings. 

I'm not sure how the review is organized. UV-absorbing compounds are discussed in Subchapter 3.5, while repair and resynthesis are discussed in Subchapter 3.6. Soon after, the authors began discussing again MAAs and scytonemin in chapter 4, which is about the molecular biology of UV-absorbing compounds. Why? Why did the authors choose to focus on this group of molecules? MAA compounds are only mentioned once in the introduction, so there is no special emphasis on these compounds. In my opinion, chapter 4 put too much emphasis on MAA in comparison to the preceding chapters and subchapters. Why is UV absorbing capacity more important than other mechanisms such as mobility, DNA repair, and antioxidant response? The authors do not justify it. I found a significant imbalance in the descriptions of the various mechanisms. Please keep in mind that the review's title refers to strategies rather than MAA. Please make the review more balanced. 

In comparison to the other chapters, even Chapter 5 (which is very interesting) is overly detailed. Chapters on nitrogen metabolism, protein content, and even DNA damage, for example, are far less informative than Chapter 5. Furthermore, chapter 5 should, in my opinion, come before chapter 3. Chapter 5 discusses how cyanobacteria detect UV radiation and initiate a response, which is likely to include tolerance and mitigation strategies. Once again, the review should be more balanced. 

Author Response

Reviewer 3

Comments and Suggestions for Authors

Issue 1: This paper is a review of the effects of UV radiation on cyanobacteria, focusing on their resilience and mitigation strategies. As a result, the review is a summary of recent findings about the processes used by cyanobacteria to tolerate high UV radiation. In general, the review is informative and precisely focused on a single subject. However, as detailed in the points below, I believe it lacks good organization. 

Response: We thank the reviewer for his kind words and taking the time to assess our MS. We have re-organized the MS to the best of our ability.

Issue 2: The abstract of the paper is sufficiently informative, but I believe it is missing one critical point, namely the need for this review article. I noticed some other reviews about UV radiation and cyanobacteria in the literature, so I asked the authors to explain why they decided to write this review, why this review is needed now, and what ultimately distinguishes this review from others. 

Response: We appreciate the reviewer’s insightful suggestions. Since, cyanobacteria are one of the primary biotic components and also a potential source of several value-added compounds therefore it is important to understand the mitigation strategies evolved with time to cope up with the drastic effects of UVR. We have revised the abstract to address your concerns and hope that the information conveyed is now clear.

Issue 3: Although I am not a native English speaker, I recommend that the authors thoroughly check the manuscript for English correctness. Perhaps a professional English speaker would be beneficial. I found several grammatical errors, such as commas between the nouns and verbs within a sentence. 

Response: We thank the reviewer for pointing this out. We have corrected the grammatical errors to the best of our ability in the revised MS.

Issue 4: What I previously mentioned for the abstract, namely the need for more in-depth justification for writing this review, also applies to the introduction. Around line 60, the authors simply stated that this review contains the most recent findings on cyanobacteria's response to UV stress, but more justification is required. 

Response: We thank the reviewer for pointing this out. We have modified the introduction section to make it more informative to the readers. Also, most recent findings on cyanobacteria's response to UV stress have been included within the revised MS.

Issue 5: The Introduction chapter is not well organized. The authors introduced the concept of programmed cell death at line 64, and they spent a few lines shortly afterwards describing the importance of PCD in the ecology of cyanobacteria. However, the authors introduced the concept of photoprotective compounds without any connection to PCD at line 70. Line 79 reiterated the scope of the review, stating that it is focused on the most recent findings, and so on. The review's scope should be presented only once. 

Response: We thank the reviewer for pointing this out. We have re-organized the introduction section and modified the text to make it more informative to the readers. The review’s scope is addressed only once in the introduction section. Please see page 2 of the revised MS, line 68-73.

Issue 6: In all honesty, I don't see the point of figure 1, at least not in its current location. Figure 1 functions as a review summary because the top section describes the effect of UV radiation on cyanobacteria at various levels. The bottom section details the mitigation strategies. This figure, in my opinion, should be divided in two. While it makes sense to show the top part at the beginning of this review, the bottom part should be included in the second part. One thing to note about the top part of figure 1: it's a little confusing. I understand the authors' preference to report on all the effects of UV radiation on cyanobacteria. However, the picture is not easily understood. What does the graph represent? I believe the image should be restyled to make it simpler. 

Response: We apologize for the inappropriate information conveyed by Figure 1. We have separated the Figure 1 into two separate figures as per your suggestion and hope that it is now clear. Also, we have restyled the first part of the figure. Please see page 3 and 8 of the revised MS, Figure 1 and 3.

Issue 7: The authors stated on line 106 that UVB radiation also affected that enzyme. What enzyme? 

Response: We have revised the text to address your concerns and hope that it is now clear. Please see page 3 of the revised MS, lines 94-100.

Issue 8: The subchapter on growth and survival is too brief to warrant its own section. It might be better to combine this with the next one about mobility. Furthermore, the sentence that UV radiation harms the growth and survival of many cyanobacteria is overly broad. Is it present at all wavelengths? Is exposure time a factor? What about UVA and UVC exposure? 

Response: We thank the reviewer for taking the time to assess our MS. We have combined the subchapters growth, survival and motility. Also, we have revised the text to address your concerns and hope that it is now clear. Please see page 3 of the revised MS, lines 115-124.

Issue 9: I noticed that the authors used the broad term UVR at times and the more specific term UVB radiation at others. I believe that they should always specify what type of UV radiation has been studied. 

Response: We appreciate the reviewer’s insightful suggestions. We have revised the text to address your concerns and hope that the information conveyed is now clear.

Issue 10: Changes in protein content are discussed in Subchapter 2.6. More information about specific proteins, in my opinion, is required. I'd like to check the article mentioned in reference 42, but the DOI link is broken, and I can't find it. 

Response: We have revised the text to address your concerns and hope that the information conveyed is now clear. Please see page 4 of the revised MS, lines 158-170.

Issue 11: What do the authors mean by "native DNA" on line 182? 

Response: We thank the reviewer for pointing this out. We have revised the text to address your concerns and hope that the information conveyed is now clear. Please see page 4 of the revised MS, lines 171-172.

Issue 12: Figure 1b, on the other hand, I believe should be placed exactly where it is first mentioned. Although the figure is sufficiently clear, the purple box contains unreadable chemical structures. 

Response: We appreciate the reviewer’s insightful suggestions. We have modified the Figure 1b to clearly indicate the chemical structure represented in the Figure. Also, the location of the Figure is modified within the revised MS as suggested. Please see page 8 of the revised MS, Figure 3.

Issue 13: There are no bibliographic citations in chapter 3.1 about avoidance.

Response: We have revised the text to address your concerns and cited the suitable references within the revised MS. Please see page 8 of the revised MS, lines 284-304.

Issue 14: Should subchapter 3.1 and 3.2 be combined? I'm wondering if avoidance is related to migration. 

Response: As suggested, we have combined the subchapter 3.1 and 3.2 within the section 4.1 in the revised MS. Please see page 8 of the revised MS, lines 282-303.

Issue 15: I dislike table 1 because it is uninformative. Molecular structures are oversized in comparison to the table, and chemical structures differ in size as well. I believe that showing the molecular structures of UV-absorbing compounds should be useful if they are related to absorbance features; otherwise, they are just drawings. 

Response: We have provided the modified table summarizing the MAAs synthesis in different cyanobacteria. Please see page 10 of the revised MS, Table 1.

Issue 16: I'm not sure how the review is organized. UV-absorbing compounds are discussed in Subchapter 3.5, while repair and resynthesis are discussed in Subchapter 3.6. Soon after, the authors began discussing again MAAs and scytonemin in chapter 4, which is about the molecular biology of UV-absorbing compounds. Why? Why did the authors choose to focus on this group of molecules? MAA compounds are only mentioned once in the introduction, so there is no special emphasis on these compounds. In my opinion, chapter 4 put too much emphasis on MAA in comparison to the preceding chapters and subchapters. Why is UV absorbing capacity more important than other mechanisms such as mobility, DNA repair, and antioxidant response? The authors do not justify it. I found a significant imbalance in the descriptions of the various mechanisms. Please keep in mind that the review's title refers to strategies rather than MAA. Please make the review more balanced. 

Response: We thank the reviewer for pointing this out. This observation is correct. We have re-organized and revised the text to address your concerns and hope that information conveyed in the revised MS is now clear.

Issue 17: In comparison to the other chapters, even Chapter 5 (which is very interesting) is overly detailed. Chapters on nitrogen metabolism, protein content, and even DNA damage, for example, are far less informative than Chapter 5. Furthermore, chapter 5 should, in my opinion, come before chapter 3. Chapter 5 discusses how cyanobacteria detect UV radiation and initiate a response, which is likely to include tolerance and mitigation strategies. Once again, the review should be more balanced. 

Response: As suggested, we have revised the text to address your concerns and tried to make the content related to subchapters “nitrogen metabolism, protein content and DNA damage” informative to the best of our ability. Also, the chapter “UV-mediated signal transduction in cyanobacteria” is placed before “UV-stress tolerance and mitigation strategies in cyanobacteria” within the revised MS. Please see page 5 of the revised MS, section 3.

Round 2

Reviewer 3 Report

I found that the authors responded to most of my previous comments more than adequately, so the paper has significantly improved over the previous version. I only have a couple of unanswered questions, mostly about the work's justification, which I believe is missing from both the abstract and introduction. 

I like how the authors changed the abstract to emphasize the importance of reviewing current knowledge about UVR's effects on cyanobacteria. However, I asked the authors to explain why they chose to write a review on the subject and how this review differs from previous ones (is it an updated version? Is the previous one out of date? Is this the first time some data has been considered and reviewed?). Justification is required... 

I should mention that the same can be said about the introduction. That section is well-written, but it lacks a strong motivation for the review and how it differs from others on the same subject. I believe it is critical to distinguish this work from others. 

Author Response

Reviewer 3

Issue 1: I found that the authors responded to most of my previous comments more than adequately, so the paper has significantly improved over the previous version. I only have a couple of unanswered questions, mostly about the work's justification, which I believe is missing from both the abstract and introduction. 

I like how the authors changed the abstract to emphasize the importance of reviewing current knowledge about UVR's effects on cyanobacteria. However, I asked the authors to explain why they chose to write a review on the subject and how this review differs from previous ones (is it an updated version? Is the previous one out of date? Is this the first time some data has been considered and reviewed?). Justification is required... 

I should mention that the same can be said about the introduction. That section is well-written, but it lacks a strong motivation for the review and how it differs from others on the same subject. I believe it is critical to distinguish this work from others. 

Response: We thank the reviewer for taking the pain and devoting time to assess our MS. We have revised the abstract and introduction part to address your concern and hope that it is now clear and acceptable to you. Please see page 1 and 2 of the revised MS, lines 26-31 and 68-76.

Previous papers related to the similar work including the work done in our laboratory report the diverse aspects of UV effects and defense mechanisms in cyanobacteria without giving much emphasis to perception of UV signals and UV-induced PCD in cyanobacteria.

In comparison to previous papers, our MS is designed to critically explain the impact of UV-radiation, UV-mediated signal perception and transduction, tolerance mechanisms and UV-induced PCD in cyanobacteria. New discoveries related to UV-absorbing compounds, PCD and repair and resynthesis are also being reported. We have tried to combine the critical topics related to the field in the form of review.